# metaVaR: Introducing metavariant species models for reference-free metagenomic-based population genomics

Romuald Laso-Jadart[1], Christophe Ambroise[2], Pierre Peterlongo[3], Mohammed-Amin Madoui[1] *

**1** Institut François Jacob, CEA, CNRS, Génomique Métabolique - UMR 8030, Univ Evry, Université Paris-Saclay, Evry, France, **2** IBGBI-LaMME, Univ Evry, Université Paris-Saclay, Evry, France, **3** Inria, CNRS, IRISA, Univ Rennes, Rennes, France

* amadoui@genoscope.cns.fr

**Data Availability Statement:** Data related to this study has been uploaded to GitHub: https://github.com/madoui/metaVaR.

## Abstract

The availability of large metagenomic data offers great opportunities for the population genomic analysis of uncultured organisms, which represent a large part of the unexplored biosphere and play a key ecological role. However, the majority of these organisms lack a reference genome or transcriptome, which constitutes a technical obstacle for classical population genomic analyses. We introduce the metavariant species (MVS) model, in which a species is represented only by intra-species nucleotide polymorphism. We designed a method combining reference-free variant calling, multiple density-based clustering and maximum-weighted independent set algorithms to cluster intra-species variants into MVSs directly from multisample metagenomic raw reads without a reference genome or read assembly. The frequencies of the MVS variants are then used to compute population genomic statistics such as $F_{ST}$, in order to estimate genomic differentiation between populations and to identify loci under natural selection. The MVS construction was tested on simulated and real metagenomic data. MVSs showed the required quality for robust population genomics and allowed an accurate estimation of genomic differentiation ($\Delta F_{ST} < 0.0001$ and $<0.03$ on simulated and real data respectively). Loci predicted under natural selection on real data were all detected by MVSs. MVSs represent a new paradigm that may simplify and enhance holistic approaches for population genomics and the evolution of microorganisms.

## Introduction

Thanks to advances in deep sequencing and metagenomics, microorganism genomic resources have become more widely available over the last two decades. By analyzing community assemblies containing a large number of uncultured species [1] we are gaining a better understanding of microbial ecology. This is especially the case for marine, soil and gut microbiomes that have been intensively investigated thanks to large sequencing consortia like *Tara* Oceans [2, 3], TerraGenome [4] or MetaHit [5].

**Funding:** The authors received no specific funding for this work.

**Competing interests:** The authors have declared that no competing interests exist.

Currently, in order to make effective use of whole-genome metagenomic data when addressing questions of molecular evolution and population genomics in uncultured species, reference genome or transcriptome sequences are required and *a priori* selected for the target species. Typically, metagenomic reads are first aligned on reference sequences for variant calling, then population alleles or amino acid frequencies [6] are obtained. Derived population genomic metrics are then computed and used to identify genomic differentiation between populations or natural selection of variants that drive genomes evolution. This approach has in particular been applied to gut microbiomes of vertebrates [7, 8] and invertebrates [9], marine bacteria [6, 10] and crustaceans [11].

In metagenomics the filtering step in which reads are aligned is critical in order to avoid the cross-mapping of reads from a given species to the reference genome of another species. The main filters are, first, the selection of genomic regions with a depth of coverage within an expected range specific to the species abundance, and, second, the minimum identity percentage of a read aligned to the reference for membership of the targeted species [11, 12].

This alignment-based approach is currently limited by the number of available reference sequences. To increase the number of references for organisms found in environmental samples, a number of methods have been developed to produce metagenome assembled species (MAGs) from whole-genome metagenomic sequencing. These approaches have successfully been applied to prokaryotes [13–15], but the main limitation of the alignment-based approach remains, namely that it is dependent on the availability, completeness and quality of reference genomes that can be constructed from metagenomic data [14]. Recently the approach has been able to make use of long-read sequencing [16, 17]. However, in the case of eukaryotes, because of the large genome size and the difficulties in obtaining high molecular weight DNA, results for eukaryotes are still lacking.

To bypass the use of references in the variant calling process, several reference-free variant calling methods have been developed. Among them, we can distinguish complete reference-free approaches as implemented in softwares like *ebwt2SNP* [18], *kmer2SNP* [19] and *DiscoSNP++* [20] to partial reference-free approaches like *Kevlar* [21] or *scalpel* [22]. These latter approaches are based on micro-assembly that generates contigs but still need a reference. None of these methods were specifically designed to deal with metagenomic data.

To identify nucleotide variants from metagenomic data, the use of *DiscoSNP++* has recently been proposed. *DiscoSNP++* detects variants by identifying bubbles in a *de Bruijn* graph built directly from the raw metagenomic reads. The variants can then be relocated on genomes of interest, if available. In comparison to alignment-based variant calling applied on metagenomic data, *DiscoSNP++* has been shown to be less sensitive but more specific in term of recall, and more accurate in term of allele frequency, especially in non-coding regions [23]. Given the sensitivity of population genomic analyses to the accuracy of the allele frequencies, we might consider the use of *DiscoSNP++* preferable to the alignment-based approach for population genomics based on metagenomic data. Moreover, the indexing method implemented in *DiscoSNP++* uses bloom filters, a space-efficient probabilistic data structure. This enables the indexing of very large and complex dataset and makes *DiscoSNP++* a very suitable tool to handle metagenomic data.

To bypass the lack of reference for micro-organisms in order to perform population genomics using metagenomic data, we first introduce the notion of metavariants, i.e variants detected directly from raw metagenomic reads without a reference genome. Then, we present an approach for clustering metavariants by species when reference genomes are not available. We propose the resulting clusters as a new form of species representation that we call *metavariant species* or MVS. We establish a formal definition of the metavariant and MVS. We implemented the clustering method in an *R* package called *metaVaR*. *metaVaR* allows the

construction of the MVSs and also their manipulation to perform population genomic analyses. The clustering of metavariants implemented in *metaVaR* was benchmark with state-of-the-art clustering algorithms. We also tested the relevance of MVSs using simulated and real metagenomic data to perform accurate population genomic analyses.

## Methods

### From metavariants to metavariant species

**Variable loci and metavariants.** We define a metavariant as a single nucleotide variant detected directly from metagenomic data without a reference genome (Fig 1). We use metavariants produced by *DiscoSNP++* and consider only metavariants located on loci producing one metavariant. Due to the absence of a reference genome, the reference ($a$) and alternative ($b$) nucleotides are chosen by *DiscoSNP++* based on alphabetic order. In a single sample, $a$ and $b$, are characterized by the count of reads supporting them, and a locus $l$ that harbors a metavariant can be represented by its depth of coverage $c$ as the sum of reads supporting $a$ and $b$. Each locus $l$ is described by the $m$ sample supporting counts $l = \{c_1, \ldots, c_m\}$. The $n$ metavariant loci row vectors $l_i$ generated from $m$ samples metagenomic data are placed in the $n * m$ depth of coverage matrix, $L = \{l_i\} = (c_{ij}) \in \mathbb{N}^{n*m}$, $i \in \{1, \ldots, n\}, j \in \{1, \ldots, m\}$. At this step, it is important to note that *DiscoSNP++* generates a fasta file of the metavariants that can be possibly mapped on a given reference.

**Clustering of metavariants into metavariant species.** A metavariant species or MVS corresponds to a set of intra-species metavariants of the same species. If it is assumed that $L$ contains both inter and intra-species metavariants, MVSs can be represented by pairwise disjoint subsets of $L$ not covering $L$.

As for the binning of metagenomic contigs [24], we consider that the depth of coverage of the variable loci of the same species covariates across samples and that this constitutes a species signature. MVSs can thus be identified by clustering $L$ based on its values. However, the complexity of metagenomic data raises several issues. First, the number of species and corresponding MVSs is unknown. Second, the initial set of metavariants contains an admixture of inter and intra-species metavariants deriving from entire genomes including repeated regions. Only

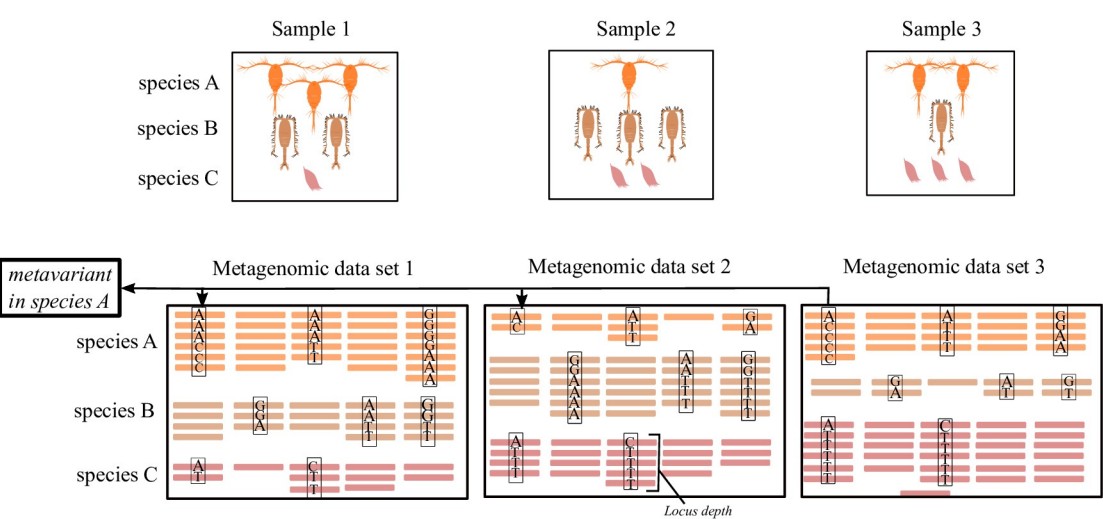

**Fig 1. Metavariants from environmental samples and metagenomic high-throughput sequencing.** The example contains three different species. Species A generates orange reads, species B brown reads and species C red reads.

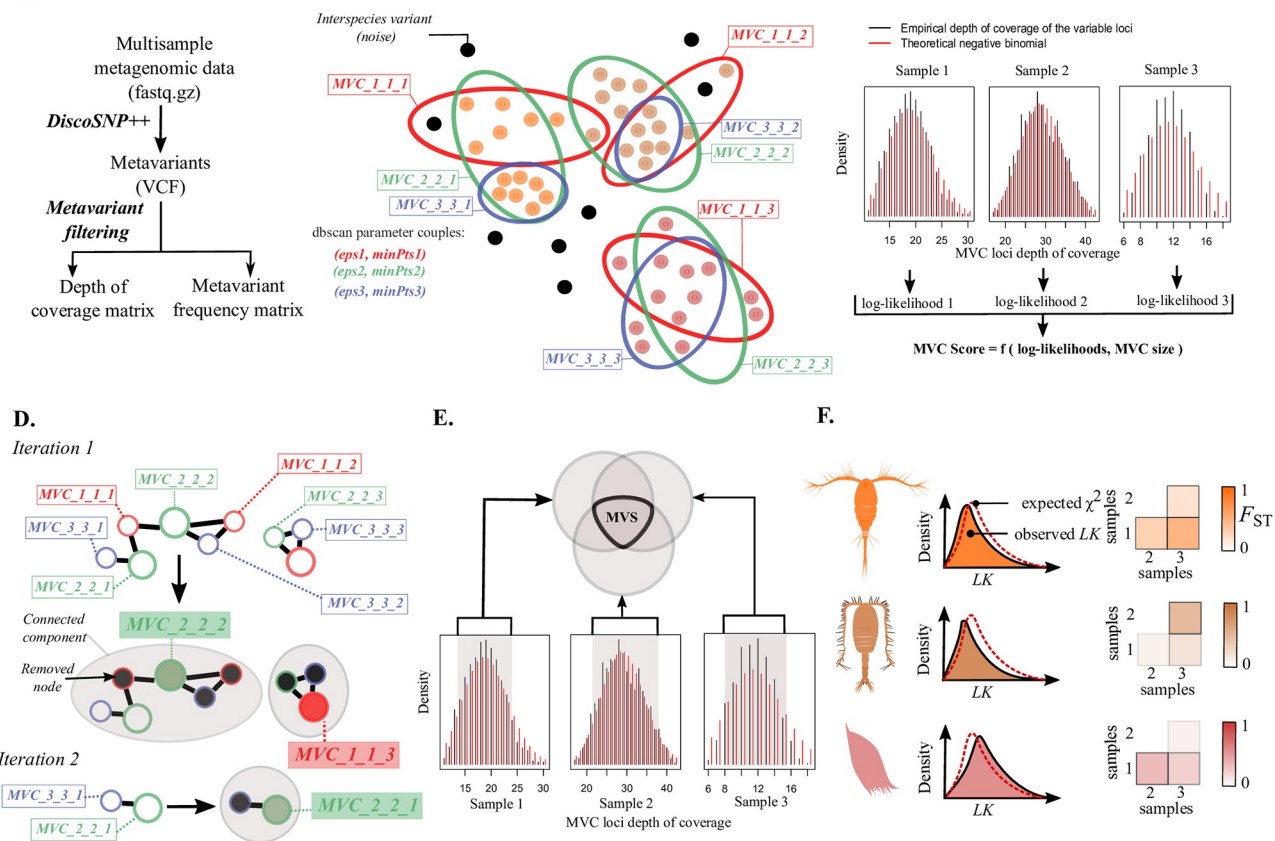

**Fig 2. Metavariant species construction from metagenomic data with metaVaR. A**. Variant calling from raw metagenomic data. **B**. Multiple density-based clustering of metavariants *mDBSCAN-WMIN*. Black points represent inter-species variants, while other colored points refer to unclustered variants. Circle colors represent the dbscan parameters. **C**. Metavariant cluster scoring. D. Maximum-Weighted Independent Set algorithm. Each node is a cluster of metavariants, their circle color representing the dbscan parameters used to build the cluster. Grey zones represent the connected components. Colored nodes are MWIS and black nodes are MWIS neighbors. **E**. Metavariant filtering for MVS construction. Grey zones correspond to the single-copy loci. **F**. Population genomics of MVS.

intra-species metavariants from single copy loci are informative for population genomics. Third, the genome size and the polymorphism rate vary considerably between species. This impacts the depth of coverage of the loci and the number of variants by species.

**Metavariant species construction steps and algorithms.** To create MVSs, we propose the following approach, described in detail in the following sections:

1. Reference-free metavariant calling with *DiscoSNP++* from raw metagenomic data (Fig 2A).

2. Metavariant filtering and construction of matrices for the depths of coverage of the different loci and for metavariant frequencies (Fig 2A).

3. Multiple density-based clustering (*mDBSCAN*) of the metavariants. Each clustering generates a set of disjoint metavariant clusters (*mvc*) (Fig 2B).

4. Each *mvc* is scored according to its size and the expected depth of coverage distribution of its loci (Fig 2C).

5. A *maximum-weighted independent set* (WMIN) algorithm is applied on all *mvc* to select a subset of *mvc* as potential MVSs (Fig 2D).

6. Selection of the metavariant loci based on coverage expectation for robust population genomic analysis (Fig 2E).

7. Computation of population genomic metrics for each MVS (Fig 2F).

**Multiple density-based clustering of metavariants.** To cluster $L$, we used density-based clustering (*dbscan*) [25]. This clustering algorithm requires two input parameters: epsilon ($\epsilon$) and minimum points ($p$), corresponding respectively to the minimum euclidean distance between two points to be considered as members of the same cluster and the minimum number of points to extend a cluster. Given $\epsilon$ and $p$, *dbscan* generates a disjoint set of metavariant clusters $\{mvc_{\epsilon,p} \subset L\}$. Intuitively, for $L$ generated from real data, there might be no optimal ($\epsilon, p$) enabling the best reconstruction of the clusters according to some criterion. Instead of choosing an arbitrary couple ($\epsilon, p$), we run *dbscan* using a grid of ($\epsilon, p$) values. This multiple density-based clustering (mDBSCAN) produces a set of possibly overlapping *mvc*. We call this set *MVC*, which by default is restricted to *mvc* containing more than 1,000 metavariants.

**Scoring of metavariant clusters.** Several metagenomic binning approaches use Gaussian or Poisson distributions to model genome sequencing coverage [13]. However, due to its overdispersion, the sequencing depth of coverage can be better approximated using a negative binomial (NB) distribution [26]. Let $mvc_{\epsilon,p,k} \in MVC$ denote the $k^{\text{th}}$ *mvc* computed with parameters ($\epsilon, p$). For each $mvc_{\epsilon,p,k}$, in each sample, we compare the observed and expected NB coverage distribution of the loci using *fitdistrplus* [27]. For all possible $mvc_{\epsilon,p,k}$ we compute $d_{\epsilon,p,k}$, the mean across $m$ samples of the log-likelihood of the fitting with $\theta_j$, the negative binomial distribution parameters in sample $j$.

$$d_{\epsilon,p,k} = \frac{1}{m}\sum_{j=1}^{m}\ln \mathcal{L}(\theta_j, mvc_{\epsilon,p,k}[j]) \tag{1}$$

with $mvc_{\epsilon,p,k}[j]$ being the depth of coverage of the metavariant in the $j^{\text{th}}$ sample in $mvc_{\epsilon,p,k}$. We also compute $\bar{d}_{\epsilon,p,k} \in [0, 1]$, the corrected mean log-likelihood of each cluster,

$$\bar{d}_{\epsilon,p,k} = \frac{d_{\epsilon,p,k} - d_{min}}{d_{max} - d_{min}}, \tag{2}$$

with $d_{min}$ and $d_{max}$ being respectively the smallest and the highest mean log-likelihood observed over all $mvc \in MVC$. We also normalised the size $s_{\epsilon,p,k}$ of each cluster such as $\bar{s}_{\epsilon,p,k} \in [0, 1]$,

$$\bar{s}_{\epsilon,p,k} = \frac{s_{\epsilon,p,k} - s_{min}}{s_{max} - s_{min}}, \tag{3}$$

with $s_{min}$ and $s_{max}$ being respectively the smallest and the highest sizes of all computed *mvc*. Finally we compute $w_{\epsilon,p,k}$, the $mvc_{\epsilon,p,k}$ score as the geometric mean between (2) and (3),

$$w_{\epsilon,p,k} = \sqrt{\bar{d}_{\epsilon,p,k} \cdot \bar{s}_{\epsilon,p,k}} \tag{4}$$

**Metavariant species as a maximum-weighted independent set.** Identifying the MVS is equivalent to simultaneously maximizing the number of non-overlapping *mvc* and their corresponding weights. This corresponds to a Maximum-Weighted Independent Set (MWIS) problem. Algorithms for this have been proposed by Sakai and colleagues [28], (Supplementary Methods 1 in S1 File). Here, we use the WMIN algorithm to find *MWIS*, the set of all MWISs.

In this context, *MVC* can be represented by a weighted undirected graph $G(V, E, W)$, where $\forall i, j \in \{1, \ldots, |V|\}$, $v_i \in V$ represents $mvc_i$ of weight $w_i \in W$ and $e_{ij} \in E \Leftrightarrow mvc_i \cap mvc_j \neq \emptyset$ and $mvc_i \neq mvc_j$. We recall here the outline of the Sakai WMIN algorithm, which takes $G$ as input and iterates until $G = \emptyset$. At each iteration the following steps are performed: (i) detection of the connected components (cc); (ii) in each cc, finding the node that is the maximum-weighted independent set, $v_i = mwis \in MWIS$, if $f(v_i) = argmax(f)$, with $f(v_i) = \frac{w_i}{deg(v_i)+1}$ and with $deg(v_i)$ the $v_i$ degree; (iii) in each cc, deletion the neighbors of the *mwis* from $G$, and storage of *mwis* in *MWIS* and deletion *mwis*. The fact that this algorithm needs $w_i \in \mathbb{R}^*$ justifies (2) and (3).

**Selection of metavariant clusters as metavariant species.** A metavariant cluster is a potential MVS if it satisfies four criteria applied in order as follows. (i) The metavariant cluster is a maximum-weighted independent set. (ii) The metavariant cluster occurs in more than $k_{min}$ populations (set to 4 by default) and corresponding loci have a median depth of coverage higher than $c_m$ (set to 8 by default). (iii) The metavariant cluster's filtered variable loci have a depth of coverage within $c_{min} = c_m - 2 * sd$, $c_{min} \geq 8$ by default and $c_{max} \leq c_m + 2 * sd$ in all samples. (iv) The metavariant cluster contains more than $m_{min2}$ metavariants (set to 100 by default). More formally,

$$mvc = mvs \in MVS \equiv mvc \in MWIS$$

$$\wedge \mid mvc \mid \geq m_{min1}$$

$$\wedge \, \forall j \in \{1, ..., k\prime\}, \ k\prime \geq k_{min}, median_{i\in[1,k]}(c_{i,j}) \geq c_m \quad (5)$$

$$\wedge \, \forall \, m_i \in mvc \equiv c_j \in [c_{min}, c_{max}], \mid \{m_j\} \mid \geq m_{min2}$$

**MVS-based population genomic analysis.** The allele frequency of metavariant species was defined as $p = \frac{c_a}{c_a+c_b}$, with $c_a$ and $c_b$ the number of read supporting the alleles $a$ and $b$. The allele frequencies are used to compute classical population genomic metrics. This includes the global $F_{ST}$ [29] such as $F_{ST} = \frac{\bar{p}}{\bar{p}(1-\bar{p})}$, with $\bar{p}$ being the mean allele frequency across all samples. We also computed the *LK*, a normalized $F_{ST}$, such as $LK = \frac{n-1}{\bar{F}_{ST}}.F_{ST}$, with $\bar{F}_{ST}$ being the mean $F_{ST}$ across all loci. *LK* is expected to follow a $\chi^2$ distribution when a large majority of the polymorphic loci are under neutral evolution [29]. To estimate the genomic differentiation between MVS populations, we compute the pairwise-$F_{ST}$ between the different populations.

## Implementation of the *mDBSCAN-WMIN* algorithm in *metaVaR*

The metavariants preprocessing step is performed by running *metaVarFilter.pl*, which produces the depth of coverage and frequency matrices from a reference-free vcf file. The *MetaVaR* package was written in R and provides three main functions for constructing MVSs:

1. *tryParam* creates metavariant clusters using several $e$, $p$ values. We used the R package *fitdistrplus* to obtain the log-likelihood of the coverage distribution.

2. *getMWIS* identify the maximum-weighted independent sets.

3. *mvc2mvs* applies filters described in (5) to select the MVSs and performs population genomic analysis.

The metaVaR source code and manual are available at https://github.com/madoui/metaVaR

## Metavariants from simulated metagenomic data

We downloaded six bacterial genomes from NCBI (*Escherichia coli* NC_000913.3, *Pseudomonas aeruginosa* NC_002516.2, *Yersinia pestis* NC_003143.1, *Rhizobium tropici* NC_020059.1, *Rhizobium phaseoli* NZ_CP013532.1, *Rhodobacter capsulatus* NC_014034.1). For each genome we created a derived genome where 1% of the genomic sites corresponds to randomly distributed SNPs for *E. coli* and 2% for the other bacteria. We used metaSim [30] to simulate Illumina paired-end 100 bp reads from 300bp genomic fragments on seven communities that contained different abundances (Fig 3A) and different proportions of original and derived genomes (Fig 3B).

To generate the metavariants, *DiscoSNP++* was run on the total read set with parameter *-b 1*. As a control, the metavariants were relocated on the six original genomes using the -G option. From the VCF file produced by *DiscoSNP++*, the depth of coverage of the biallelic loci and allele frequencies were calculated using *metaVarFilter.pl* with parameters *-a 10 -b 500 -c 7*. Here, the first two parameters are the minimum and maximum cumulative depths of coverage of a locus, and the third parameter specifies that a locus is kept only if it occurs in at least seven samples.

## Metavariants from real metagenomic data

To test the performance of the *mDBSCAN-WMIN* algorithm on real metagenomic data, we used metagenomic reads from five marine samples collected in the Mediterranean (Table 1). In a previous study, the reads were processed by *DiscoSNP++* and the metavariants were aligned on the *Oithona nana* genome. Then, the genomic differentiation between samples was estimated by pairwise-$F_{ST}$ and loci under selection were identified [23].

In the present study, *DiscoSNP++* was run on the five read sets and the vcf output was filtered using *metaVarFilter.pl* with parameters *-c 20 -b 250 -c 3*. This produced two files containing the depth of coverage of metavariant loci and the metavariant frequency matrices. These two files were then used as input for *metaVaR* and other algorithms used for benchmarking (see next section for details).

As a control, the metavariants belonging to *O. nana* were identified by mapping the metavariants back onto the *O. nana* genome. The *MWIS* corresponding to *O. nana* was used to compare the genomic differentiation (pairwise-$F_{ST}$) estimated by *metaVaR* to the expected pairwise-$F_{ST}$ values computed by the reference-based approach.

In *O. nana*, loci with *LK* values that were higher than expected (based on the $\chi^2$ distribution) were considered under selection for *p*-value$\leq 0.05$.

## Comparison of *mDBSCAN-WMIN* to other sequence abundance-based clustering algorithms

The depth of coverage matrix was used for metavariant clustering by *mDBSCAN-WMIN* using parameters $e = (3, 4, 5, 6, 7)$ and $p = (5, 8, 10, 12, 15, 20)$. There are no clustering algorithms explicitly developed for metavariant clustering. It is though possible to solve this clustering problem using other sequence abundance-based clustering algorithms, such as those used in RNA-seq data analysis to identify co-expressed genes. We tested state-of-the-art clustering algorithms as follows using *coseq* [31]: (i) centered log-ratio transform and k-means clustering (with k values from 2 to 12); (ii) arcsin transform and Gaussian Mixture Model with the same k values as in (i); (iii) logit transform and GMM with the same k values as in (i).

To evaluate the performances of each clustering algorithm on simulated data, the clusters were first assigned to one single original genome. It was based on the highest proportion of

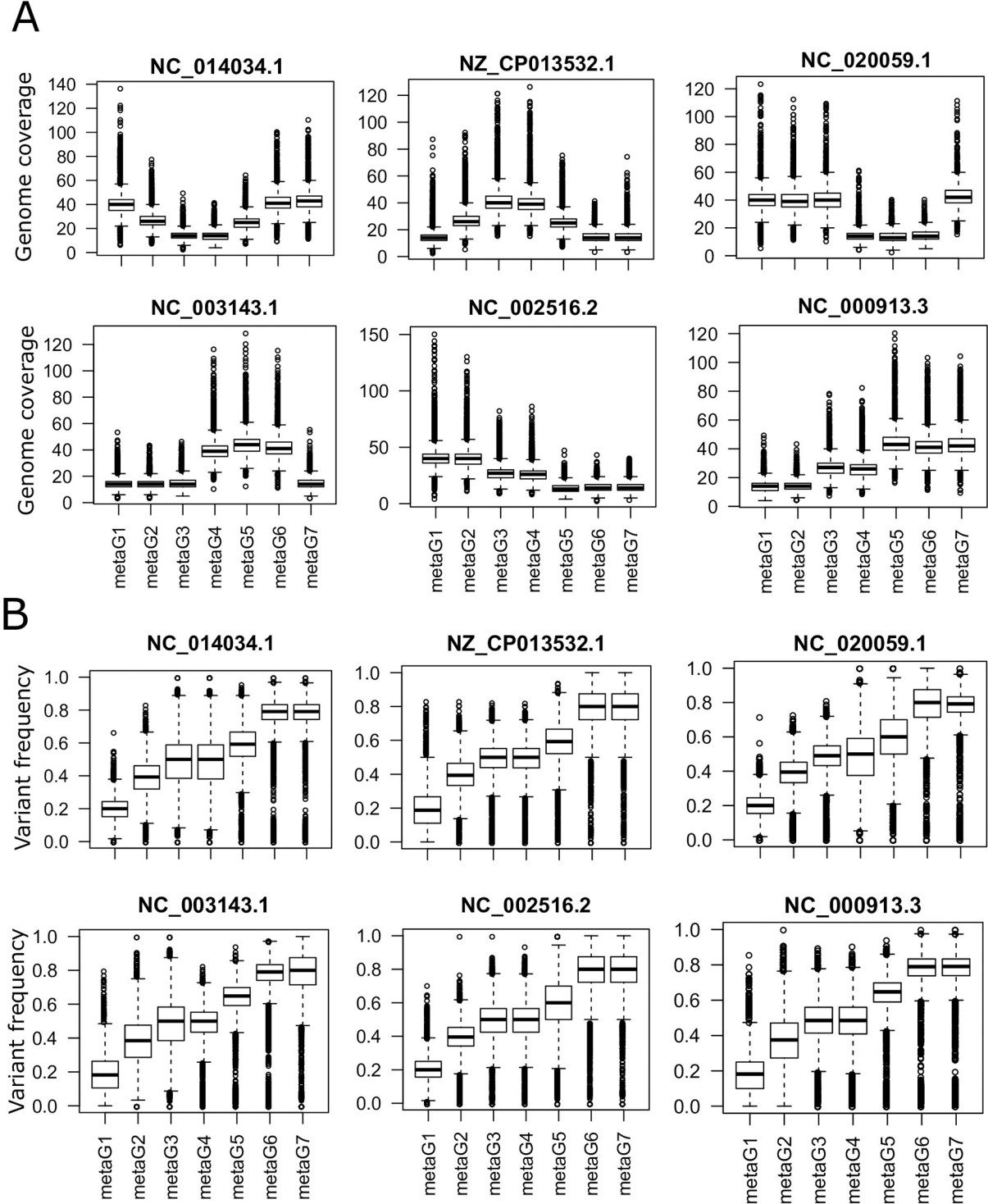

**Fig 3. Simulated seven-metagenomic dataset on an admixture of six bacterial species containing within-species single nucleotide polymorphism.** A. Variable loci genomic coverage distribution of the species. B. Within-species variants frequencies distribution.

**Table 1. Metavariant clustering performances on simulated metagenomic data.**

| Algorithm | Software | Recall | Precision | Signal-To-Noise | Purity | Entropy | Time CPU (min)* |
|---|---|---|---|---|---|---|---|
| mDBSCAN-WMIN | metaVaR | 0.5523 | **0.9996** | **1691.18** | **0.9999** | **0.008** | 3.96 |
| CLR-kmeans | coseq | **0.5945** | 0.9941 | 99.79 | 0.993 | 0.22 | **1.66** |
| arcsin-GMM | coseq | 0.3007 | 0.998 | 158.24 | 0.9915 | 1.77 | 4.41 |
| logit-GMM | coseq | 0.2685 | 0.9993 | 408.6 | 0.9892 | 1.76 | 5.94 |

*The computation was performed on Intel(R) Core(TM) i5-7200U CPU.

metavariants originating from the same genome, using the results of the alignment of the metavariants performed with *bwa mem* with default parameters.

For a cluster, we defined the true positives *TP* as the number of metavariants of the cluster deriving from the original genome, the false positives *FP* as the number of metavariants of the cluster not deriving from the original genome, the false negatives *FN* as the number of metavariants of the original genome not present in the cluster. We computed the recall, precision and signal to noise (STN) of each cluster, as follows: $recall = \frac{TP}{TP+FN}$, $precision = \frac{TP}{TP+FP}$, $STN = \frac{recall}{1-precision}$, with *TP*, the number true positives, *FN* the number of false negatives and *FP* the number of false positives.

The global purity and entropy of the clustering was computed as follows: $Purity = \frac{1}{n}\sum_{q=1}^{k} \max_{1\leq j\leq l} n_q^j$ with *n* the number of metavariants, $n_q^j$ the number of metavariants in cluster *q* belonging to original species *j*, and $Entropy = -\sum_{q=1}^{k}\sum_{j=1}^{l} \frac{n_q^j}{n_q} \log_2 \frac{n_q^j}{n_q}$, where *n* is the total number of metavariants, $n_q$ the total number of samples in cluster *q*, and $n_q^j$ the number of samples in cluster *q* belonging to the original species *j*.

To evaluate the accuracy of the (pairwise-$F_{ST}$) computed from the simulated data by *metaVaR*, we calculated the difference between all possible (pairwise-$F_{ST}$) for each bacteria computed by *metaVaR* to the expected pairwise-$F_{ST}$ obtained with the alignment-based approach according to the pattern presented on Fig 3B where, for each bacteria, the abundance of the derived genome increases continuously from sample metaG1 to sample metaG7.

To test the sample size effect on the metaVaR performances, we sampled all possible combinations of three to six metagenomes from the seven initially simulated metagenomes. Then, we ran metaVaR using the following grid parameters *e* = (3, 4, 5, 6, 7) and *p* = (5, 8, 10, 12, 15, 20). For each run, we calculated the purity, entropy, precision and recall as previously described.

## Results

### Metavariant species as a new modelling of organisms from metagenomic data

In the absence of a reference genome to guide metagenomic data analyses for population genomics, we model species only by their variable loci. They are characterized by their associated depths of coverage and variant frequencies across environmental samples. We called this model *metavariant species* or MVS, and we proposed a method for constructing MVSs from multisample raw metagenomic data (Fig 1). The method is based on reference-free variant calling using metagenomic reads from different samples by *DiscoSNP++* (Fig 2A).

In the context of metagenomics, the variants are termed metavariants and clustered into MVSs. The metavariants are clustered by multiple density-based clustering (*mDBSCAN*) based

on the covariation of the depth of coverage of the corresponding variable loci across samples (Fig 2B). Clusters are then scored statistically based on the expected depth of coverage of the variable loci they contain in each sample (Fig 2C). The best clusters are selected by a maximum-weighted independent set (WMIN) algorithm (Fig 2D) and variable loci with a minimal coverage threshold are selected to obtain the final MVSs (Fig 2E). The method was implemented in a *R* package called *metaVaR* that allows users to build and manipulate MVSs for population genomic analyses (Fig 2F). The relevance of the MVSs and *metaVaR* was tested for population genomics use.

## Metavariant species on simulated metagenomic data

To test the relevance of MVSs for population genomic analyses, we simulated seven metagenomic data sets composed of Illumina short reads from an admixture of six bacteria in various abundances (Fig 3A). Each bacterial species was composed of two strains in various abundances (Fig 3B), with a continuous increase of the derived strains abundance from metaG1 to metaG7. Metavariants were detected by *DiscoSNP++* and filtered, giving 90,593 metavariants from which the depths of coverage of the variable loci and metavariant frequency matrices were computed.

The metavariants were clustered into MVS candidates using the *mDBSCAN-WMIN* algorithm, and the clustering performances were compared with the performances of three state-of-the-art algorithms [31]: (i) centered log-ratio and kmeans (CLR-kmeans); (ii) arcsin transform + Gaussian mixture model (arcsin-GMM); (iii) logit transform + Gaussian mixture model (logit-GMM).

The clustering performances of the four algorithms are summarized in Table 1 and illustrated in Fig 4. Overall, the *mDBSCAN-WMIN* algorithm had the highest precision, signal to noise ratio (STN), purity and entropy. *mDBSCAN-WMIN* was slightly but not significantly less sensitive than *CLR-kmeans* (Fig 4A), (paired U-test, $P \geq 0.05$). *mDBSCAN-WMIN* was significantly more precise than the three other algorithms (Fig 4B) and had a significantly higher STN (Fig 4C) (paired U-test, $P \leq 0.05$). Moreover, three of the *mDBSCAN-WMIN* clusters out

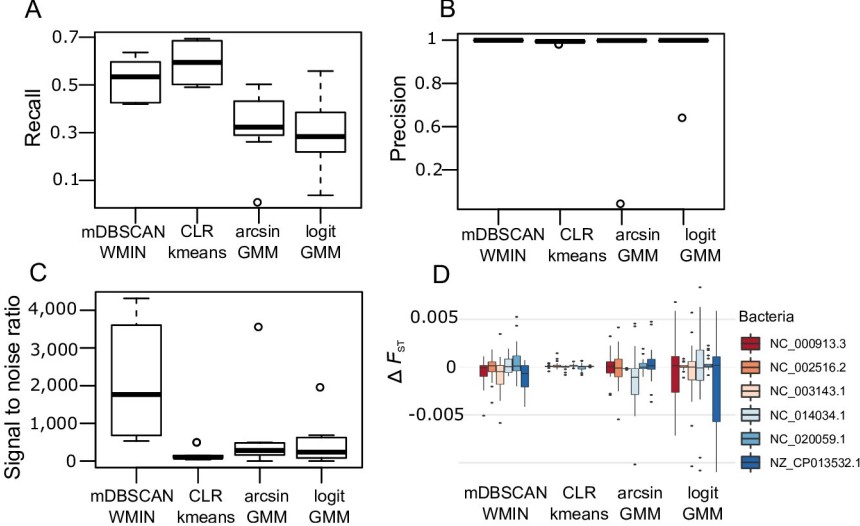

**Fig 4. Metavariant clustering performances of four clustering algorithms.** A. Recall of metavariant clusters. B. Precision of metavariant clusters. C. Signal to noise of metavariant clusters. D. Population pairwise-$F_{ST}$ difference between MVS and the alignment-based method.

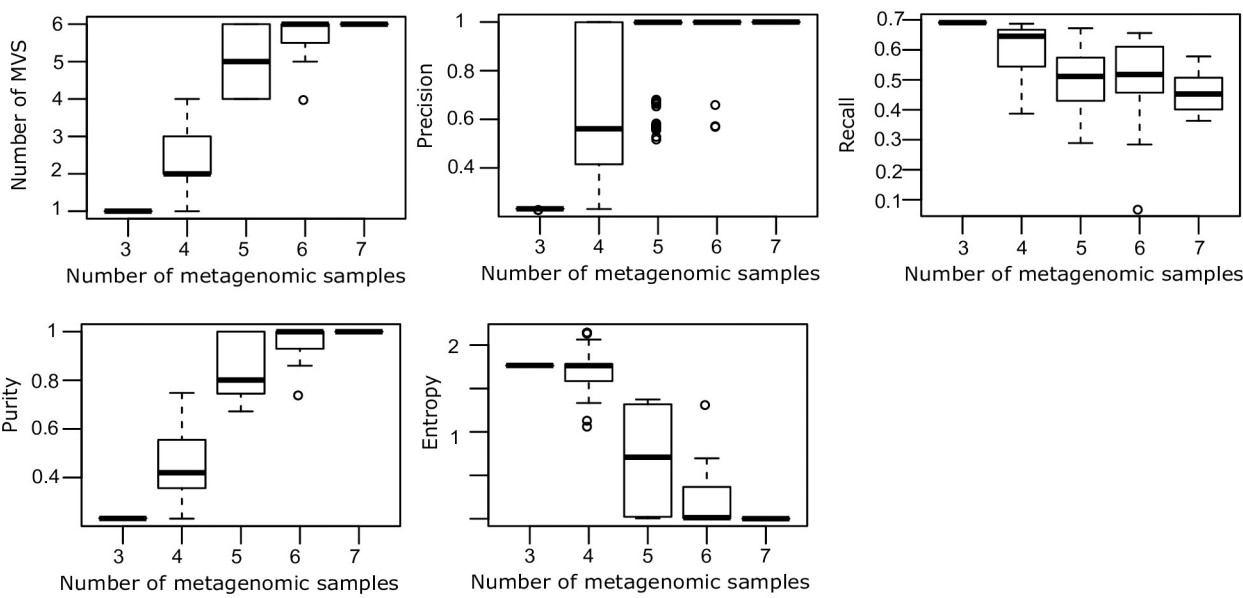

**Fig 5. Sample size effect on metavariant clustering performances.**

of the six contained zero false positives. The arcsin-GMM and logit-GMM methods showed significantly lower recall (paired U-test, $P \leq 0.05$).

The six clusters selected by *mDBSCAN-WMIN* correspond to three different *DBSCAN* parameter settings. Four clusters generated with $e = 6$, $p = 8$ were selected, one cluster for $e = 6$, $p = 5$ and one for $e = 7$, $p = 12$. This is a good illustration that with our scoring method there is not one single couple $(e, p)$ that gives the highest-scoring metavariant clusters.

The samples size effect on the metaVaR performances was tested by sampling several times different number of metagenomes (Fig 5). We observed that the precision and purity increase with the number of metagenomes used as input while the recall and entropy decrease. The MVSs corresponding to the six initial species can be found using a minimum of five metagenomic samples.

The accuracy of the pairwise-$F_{ST}$ estimation based on MVS was evaluated for the four clustering algorithms and compared with the accuracy obtained using metavariant alignment on the six bacterial genomes (Fig 4D). Pairwise-$F_{ST}$ estimated on clusters from the four algorithms showed negligible differences with the reference-based approach for $\Delta_{F_{ST}} \leq 0.01$. However, the *CLR-kmeans* algorithm showed the lowest $\Delta_{F_{ST}}$ values.

## Metavariant species on real metagenomic data

To evaluate the relevance of MVS on real metagenomic data generated from environmental samples containing more complex genomes than the bacterial samples, we selected five metagenomic marine samples known to contain the zooplankter *Oithona nana* in sufficient abundance for population genomic analyses [23].

We ran *DiscoSNP++* on the raw data and generated 1,159,157 metavariants, filtered into 138,676 metavariants. The metavariants were clustered into MVSs using the same four clustering algorithms as previously tested for simulated data. MVSs corresponding to *O. nana* were identified by aligning the metavariants on its genome. *mDBSCAN-WMIN* and *CLR-kmeans* detected the *O. nana* MVS, but the two other methods generated the maximum number of MVSs allowed by the parameters (i.e 12 clusters), with no clusters assigned to *O. nana*. These

**Table 2. Metavariant clustering performances on real metagenomic data.**

| Algorithm | Software | Number of cluster | Recall* | Precision* | Signal-To-Noise* | Time CPU |
|---|---|---|---|---|---|---|
| mDBSCAN-WMIN | metaVaR | 3 | 0.1592 | **0.8144** | **1691.18** | **1.48** |
| CLR-kmeans | coseq | 4 | **0.1662** | 0.7182. | 99.79 | 4.5 |
| arcsin-GMM | coseq | 12 | - | - | - | 5.53 |
| logit-GMM | coseq | 12 | - | - | - | 4.9 |

*Performances for the *O. nana* cluster. Time CPU is in minutes.

two other methods are therefore not considered further (Table 2). The *O. nana* MVS built by *mDBSCAN-WMIN* is less complete but more accurate than that built by *CLR-kmeans* (Table 2). The metavariants that were not relocated on the *O. nana* may include missing parts of the genome assembly.

The pairwise-$F_{ST}$ matrices of the *O. nana* MVSs (Fig 6A) showed small differences in relation to alignment-based $F_{ST}$ values ($\Delta_{F_{ST}} \leq 0.03$) (Fig 6B). To illustrate potential MVS applications, we performed several downstream analyses, including isolation-by-distance (IBD) (Fig 6C), species co-differentiation (Fig 6D), and natural selection tests (Fig 6E and 6F). In the Mediterranean, the Lagrangian distances between the western and eastern basin sampling sites (S10, 11, 12 and S24, 26 respectively) can to a large extent explain the *O. nana* genomic differentiation (Mantel $r = 0.73$, $p - value \leq 0.05$) (Fig 6C) [32]. Comparing the genomic differentiation trends between three MVSs (detected by the *mDBSCAN-WMIN*) revealed a negative correlation between MVS2 and MVS3, but no co-differentiation patterns between other MVS pairs.

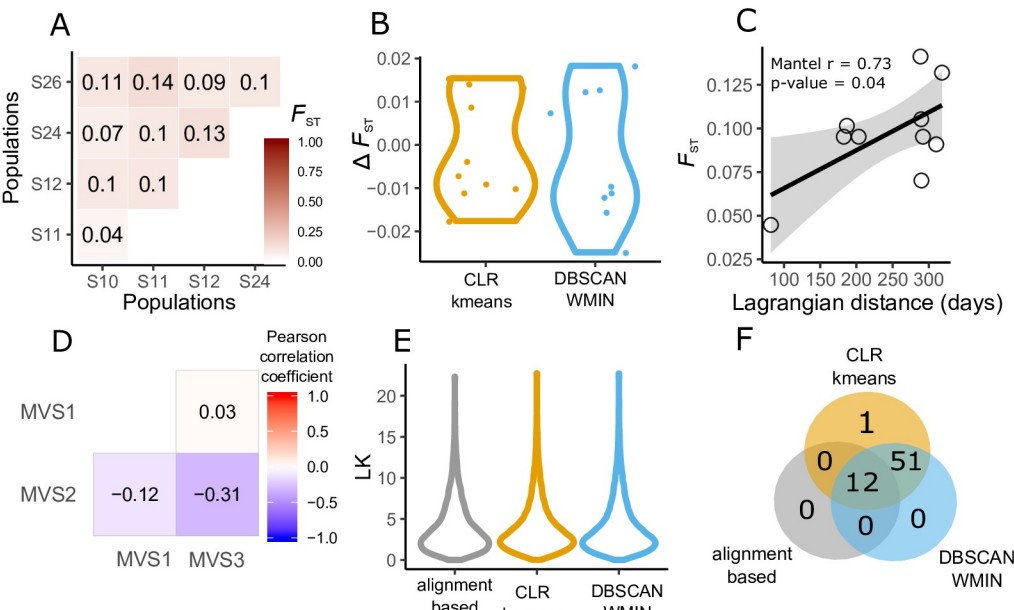

**Fig 6. Examples of applications and accuracy of metavariant species. A**. Genomic differentiation based on pairwise $F_{ST}$ of *O. nana* populations from the MVS built with *DBSCAN-WMIN*. **B**. Difference of $F_{ST}$ values between alignment-based method and MVSs. **C**. Mantel test with Lagrangian distances. **D**. Co-differentiation between species. **E**. LK distribution. **F**. Venn diagram of loci predicted under selection (LK p-value $\leq 0.001$) by the different approaches.

Loci under selection in Mediterranean populations of *O. nana* were identified based on LK outlier values produced by the MVS-based and the alignment-based methods. For all three approaches, the LK distribution suggests neutral evolution at most of the *O. nana* polymorphic sites (Fig 6E). The prediction of loci under selection based on extreme LK values in the three approaches showed that all loci predicted under selection by the alignment-based approach are identified in the *O. nana* MVSs. Plus, more loci are identified under selection by both clustering methods, which differs only for one loci (Fig 6F).

Loci characterized by extreme LK values were positively detected with the three approaches. The 12 loci predicted under selection by the alignment-based approach are identified in the *O. nana* MVSs. More loci are identified under selection by both clustering methods, which differs only for one loci (Fig 6F).

## Discussion

### Modeling species' nucleotide polymorphism by metavariant species

Large-scale nucleotide polymorphism detection traditionally requires reference sequences such as a genome or transcriptome assemblies. In most species, these resources are lacking, which greatly reduces the scope of population genomic investigations, since the choice of species is limited to those that have been sequenced. The small number of species that have a transcriptome or genome reference are unrepresentative of the whole genomic landscape of small eukaryote-rich biomes found in the oceans in particular [2]. MVS representation allows this obstacle to be overcome, making it possible to investigate a much larger number of species, including unknown species for which few or no genomic resources are available. However, if a reference sequence is available for a targeted species, we recommend also to use classical reference and alignment-based approaches.

MVS modelling nevertheless still requires a minimal amount of genomic information, which includes the variable loci of single copy regions and their variant frequencies in different samples. This information is sufficient for basic population genomic analyses, such as genomic differentiation and detection of loci under natural selection. The extraction of this information from raw metagenomic data through the use of *DiscoSNP++* does not need a reference or assembly, and generates accurate variant frequencies in a reasonable time using computational resources, even for very large datasets [23].

Our results showed a large amount of false negative metavariants that where not assigned to any MVS using both *mDBSCAN-WMIN* or other *coseq* clustering algorithms. This can be explained by the fact that we used only the metagenomic abundance. The reads depth of coverage distribution is known to be over-dispersed when using Illumina sequencing [26], which disables to cluster all loci into MVC in the simulated data and *a fortiori* in real data. However, we proved that the accuracy of the population genomic metrics is not affected by this false negative rate (or low recall) and *metaVaR* estimates accurate $F_{ST}$ with both simulated and real data.

To ease the MVS-based population genomic analysis, we selected only the biallelic loci which *defacto* reduces the complexity of the analysis but allows to compute directly the allele frequency instead of working on the four nucleotides frequencies.

As the allele frequency estimation bias depends greatly on the depth of coverage. High depth of coverage (>20x) allows more accurate allele frequency estimation, but such coverage are not always reached in metagenomic samples. However, here the selection of loci with at least 8x is sufficient to provide useful information for accurate allele frequency estimation.

In order to gain an ecological insight from MVSs, we recommend performing a taxonomic assignment of the MVSs, which can be done by aligning the variable loci sequences generated

by *DiscoSNP++* against public sequence databases. Due to their short length, metavariants can be aligned using classical short read aligners. Such taxonomic assignment of metavariants has already been successfully performed on real large metagenomic data set [33]. Moreover, for users with specific interest in a particular species, genome or transcriptome references can be used directly during the variant calling step using only *DiscoSNP++* with no need of *metaVaR*.

## The challenge of metavariant classification by species

The accuracy of the genomic differentiation estimation depends on the sample size and the number of markers. Having an exhaustive set of SNPs is not mandatory, but a large set ($> 1,000$ SNPs) is preferable [34]. The number of metavariants required for an MVS to be considered valid is critical. Moreover, the loci under selection often represent a small fraction of the genome. Increasing the number of metavariants in an MVS can help to detect these loci, but it is crucial to avoid false positives that generate biased $F_{ST}$ values. In metagenomics, false positives are metavariants assigned to a species that they do not belong to. $F_{ST}$ values derived from mis-assigned metavariants may generate outliers and support false signals of natural selection. For this reason precision may be deemed to be the criterion with the highest priority. Clustering the metavariants using *mDBSCAN-WMIN* and *CLR-kmeans* gave the best clustering results on simulated and real data, but *mDBSCAN-WMIN* is nonetheless more specific but less sensitive. Both methods have also their own particular limitations. With *mDBSCAN-W-MIN* a variety of clustering parameters ($e$, $p$) generally needs to be tried. For example, values for $e$ ranging from 0.1 to 1 and $p$ from 10 to 100 can be initially tried. However, several runs will often be necessary to obtain all possible MVSs. *CLR-kmeans* involves trying different values of $k$ with no prior knowledge, and the optimum value of $k$ may be missed.

From our simulations, the number of metagenomic samples appeared to be critical to find all the possible MVSs and it seems reasonable to use at least six metagenomic samples to obtain MVSs with a high precision. However, the between samples homogeneity of the species relative abundance may also affect the performance of the method. Meaning that two species whose abundance covariates may be clustered in the same MVS. To cope with this problem, increasing the number of metagenomic samples remains a good option. However, as demonstrated by our simulations and due to metavariants filtering, the increasing number of samples will also lead to a decrease of the recall.

## Toward a holistic view of microorganism genomic differentiation and natural selection

Current population genomic analyses focus on one single species at a time for which we do not have a sequence reference. Thanks to MVSs, the genomic differentiation of several species without genomic reference can be modelled simultaneously, and hypotheses like isolation-by-distance can be tested on each species. The genomic differentiation of MVSs can be compared and species sharing common differentiation profiles can be identified to illustrate possible co-differentiation or similar gene flow. However, even if the method is able to retrieve several MVSs from multiple samples, this will never cover the whole complexity within and among samples. Indeed, only few species present a sufficient sequencing depth of coverage and enough metavariants to pass the quality filters in real samples.

To give an idea of the potential of metaVaR on real large metagenomic data, we tested our method on four data sets containing several millions of metavariants from a total of 114 samples produced in a previous study [23] using *Tara* Oceans data and we obtained a total of 113 MVSs (Supplementary Result 1 in S1 File). The investigations of these MVSs will hopefully lead to new knowledge on plankton molecular evolution.

Another relevant MVS application concerns natural selection. The ratio of loci under selection over the total number of variable loci is an interesting metric for estimating the impact of natural selection on the molecular evolution of a species. This ratio can be computed for each MVS and the different ratios compared in order to assess the relative effect of natural selection.

## Current limitations and future developments for metavariant species-based population genomics

Pairwise-$F_{ST}$ currently remains a robust metric for tracing the silhouette of the genomic differentiation from metagenomic data. The absence of genotypes and haplotypes and their relative frequencies precludes intra-population analysis, and makes it impossible to compute p-values for $F_{ST}$. Moreover, the use of population genomic tools enabling the estimation of nucleotide diversity, the identification of genomic structure, and the testing of evolutionary trajectories and past demographic events is not yet possible. For these reasons, future developments focusing on variant phasing and haplotyping from metagenomic data will greatly help to improve MVS applications. In this context, the use of long read sequencing technologies will be of great benefit, by supporting long-range haplotypes spanning several kilobases.

## Conclusion

MVSs make it feasible to carry out population genomic analyses of unknown organisms without a reference genome or genome assembly. MVSs are suitable for genomic differentiation and natural selection analysis. Simultaneous access to nucleotide polymorphisms of different species present in the same ecosystem allows for a holistic view of microorganism genomic differentiation and adaptation. Future developments will attempt to reconstruct species haplotypes based on metavariant species, in order to provide a more accurate view of species evolution.

## Supporting information

**S1 File.**
(PDF)

## Acknowledgments

Our thanks to David Vallenet, Guillaume Gautreau and Adelme Bazin for helpful discussions.

## Author Contributions

**Conceptualization:** Mohammed-Amin Madoui.

**Investigation:** Romuald Laso-Jadart.

**Methodology:** Romuald Laso-Jadart, Christophe Ambroise, Pierre Peterlongo, Mohammed-Amin Madoui.

**Supervision:** Mohammed-Amin Madoui.

**Validation:** Mohammed-Amin Madoui.

**Writing – original draft:** Mohammed-Amin Madoui.

**Writing – review & editing:** Romuald Laso-Jadart, Christophe Ambroise, Pierre Peterlongo, Mohammed-Amin Madoui.

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
