## [Decision Letter · Decision Letter 0]

13 Nov 2020

PONE-D-20-25895

metaVaR: introducing metavariant species models for reference-free metagenomic-based population genomics

PLOS ONE

Dear Dr. Madoui,

First of all apologies for the long delay in assessing your manuscript. It is hard to find reviewers for bioinformatics manuscripts. Thank you for submitting your manuscript to PLOS ONE. After careful consideration, we feel that it has merit but does not fully meet PLOS ONE’s publication criteria as it currently stands. Therefore, we invite you to submit a revised version of the manuscript that addresses the points raised during the review process.

We look forward to receiving your revised manuscript.

Kind regards,

Francisco Rodriguez-Valera

Academic Editor

PLOS ONE

Journal Requirements:

2. Please amend your list of authors on the manuscript to ensure that each author is linked to an affiliation. Authors’ affiliations should reflect the institution where the work was done (if authors moved subsequently, you can also list the new affiliation stating “current affiliation:….” as necessary).

Reviewers' comments:

Reviewer's Responses to Questions

**Comments to the Author**

1. Is the manuscript technically sound, and do the data support the conclusions?

Reviewer #1: Yes

Reviewer #2: Yes

2. Has the statistical analysis been performed appropriately and rigorously? 

Reviewer #1: No

Reviewer #2: Yes

3. Have the authors made all data underlying the findings in their manuscript fully available?

Reviewer #1: Yes

Reviewer #2: Yes

4. Is the manuscript presented in an intelligible fashion and written in standard English?

Reviewer #1: No

Reviewer #2: Yes

5. Review Comments to the Author

Reviewer #1: In the article "metaVAR: introducing metavariant species models for reference-free metagenomic-based population genomics", the authors present a method to retrieve polymorphism data from metagenome samples without the need for a reference genome, based on performing variant calling directly on the raw reads, then grouping the different variants into "Metavariant Species" (MVS) based on the variant coverage.

The method is mathematically sound and it certainly has a place in current metagenomic analyses. Although the production of Metagenome Assembled Genomes (MAGs) is widespread on the field, there are still a lot of samples for which the assembly of contigs to build MAGs is still difficult (One such example is any sample where one species of microorganism is specially abundant, such as salterns). However, I see a few issues that need to be addressed before this article can be published.

One of them is the overall readability of the article, specially the introduction. Although there are not any grammatical or spelling errors, there are some phrases that should be rewritten to make it easier to understand. I would bring special attention to lines 11-16, 50-53 and 256-275, but the entire article could do with an additional punctuation pass. The figures could do with a bit of retouching: Figure 1 in particular has too much information, including pictures referring to different sections of the article. Offloading some of the subfigures to Supplementary data would make the figure easier to parse. I am also missing a figure explaining in a graphical manner the concept of variable loci and metavariants: understanding these concepts is vital to understand the method, but the definition in the text is correct but a tad too formal for someone without proper mathematics formation, and a good figure could help biologists understand the core concepts of the article. The Algorithm 1 could also be removed from the article, as it is not required to understand the method and could be moved to Supplementary Data.

My main gripe with the article, however, is the application of this method to real metagenomic analyses. I'm particularly worried about the following issues:

*The scoring of each metavariant cluster depends on the number of samples, but metagenomic studies do not usually include replicates and the amount of samples sequenced is usually very low. How does sample size affect metavariant clustering? How many samples are needed to obtain a good separation of clusters? Does the homogeneity of the samples (How similar are they to one another, in terms of population composition) affect the scoring somehow?

* I am particularly troubled about the number of metavariants the method is able to recover: the simulated metagenomic dataset test only uses 6 genomes, and the real metagenomic test focuses only on a single genome. A real metagenomic sample is going to have a lot more genomes. How many is the tool able to recover? If the tool is only able to recover a limited number of genomes per run, is it possible to direct the tool to recover an specific genome?

* Although I understand the motivation of building MVS, a biologist using this tool still needs a way to connect a MVS to a genome, which the article does not include.

If these issues are adressed, then I have no doubt metaVaR has a bright future as a tool in population genomics.

Reviewer #2: In this manuscript the authors introduce a new method to detecting micro-diversity in metagenomic datasets. They also introduce the concept of metavariants and metavariant species. Their tool, MetaVaR was tested on both real and simulated datasets and presented superior performance. The manuscript is relevant and well written, the experiments are well designed and the results support the conclusions. Nevertheless some minor changes and clarifications are necessary to make this manuscript suitable for publication.

Ln 71: It seems to me that considering only a single metavariant might be a major limitation of the proposed method, albeit necessary to make the computations feasible. This should be mentioned in the discussion.

Ln 73: Would it not be more logical to establish the reference as the variant that has higher coverage?

Ln 113: Which distance metric was used with the DBSCAN algorithm? Did you test different metrics?

Ln 189-191: This setup is an overly simplistic representation of metagenomes. In most real world datasets metagenomes are made up of many more species and often multiple strains of the same species. Using only six species and no multiple strains from the same species is likely to inflate the precision and skew the other evaluation metrics.

Ln 195: Perhaps the term “communities” is more adequate than “populations” considering you are dealing with different species.

Figure 1C: Is there evidence to support that the DBSCAN noise points are all the result of inter-species variation or is this one of your assumptions? In case of the latter, this should be explicitly stated along with the rationale behind it.

Figure 2A: Is it correct to assume that coverage values refer to specific positions within genomes rather than coverage of the whole genome? If so, please clarify

Ln 253: word missing after evolutionary?

6. PLOS authors have the option to publish the peer review history of their article (what does this mean?). If published, this will include your full peer review and any attached files.

Reviewer #1: No

Reviewer #2: No

---

## [Author Response · Author response to Decision Letter 0]

3 Dec 2020

Dear editor and reviewers,

I really appreciate the time spent on reviewing our study. We took all your remark in consideration and redacted a complete response point by point to satify your legitimate concerns. We hope the effecort we provided fill fill your expectations. The manuscript contains major changes and new analysis and we hope the readers will find a great interest on our work. A detailed answer is provided in a seperate file.

Regards 

Best regards,

The authors

---

## [Editor Report · Decision Letter 1]

15 Dec 2020

metaVaR: introducing metavariant species models for reference-free metagenomic-based population genomics

PONE-D-20-25895R1

Dear Dr. Madoui,

We’re pleased to inform you that your manuscript has been judged scientifically suitable for publication and will be formally accepted for publication once it meets all outstanding technical requirements.

Kind regards,

Francisco Rodriguez-Valera

Academic Editor

PLOS ONE

Additional Editor Comments (optional):

Sorry for the delays